# On-surface synthesis and spontaneous segregation of conjugated tetraphenylethylene macrocycles

En Li[1], Cheng-Kun Lyu[1], Chengyi Chen[1], Huilin Xie[2], Jianyu Zhang [2], Jacky Wing Yip Lam[2], Ben Zhong Tang[2,3] & Nian Lin [1✉]

Creating conjugated macrocycles has attracted extensive research interest because their unique chemical and physical properties, such as conformational flexibility, intrinsic inner cavities and aromaticity/antiaromaticity, make these systems appealing building blocks for functional supramolecular materials. Here, we report the synthesis of four-, six- and eight-membered tetraphenylethylene (TPE)-based macrocycles on Ag(111) via on-surface Ullmann coupling reactions. The as-synthesized macrocycles are spontaneously segregated on the surface and self-assemble as large-area two-dimensional mono-component supramolecular crystals, as characterized by scanning tunneling microscopy (STM). We propose that the synthesis benefits from the conformational flexibility of the TPE backbone in distinctive multistep reaction pathways. This study opens up opportunities for exploring the photophysical properties of TPE-based macrocycles.

[1] Department of Physics, The Hong Kong University of Science and Technology, Clear Water Bay, Hong Kong, China. [2] Department of Chemistry, Hong Kong Branch of Chinese National Engineering Research Center for Tissue Restoration and Reconstruction, The Hong Kong University of Science and Technology, Clear Water Bay, Hong Kong, China. [3] School of Science and Engineering, Shenzhen Institute of Aggregate Science and Technology, The Chinese University of Hong Kong, Shenzhen, Guangdong, China. ✉email: phnlin@ust.hk

Since the discovery of macrocyclic crown-ethers by Pedersen more than half a century ago[1], macrocycles have become versatile building blocks in supramolecular chemistry[2]. Specifically, inner cavities of shape-persistent macrocycles can host molecules or ions forming host-guest supramolecular complexes[3,4]. In contrast to the open-chain oligomers, the cyclic topology gives rise to unique electronic and optical properties such as aromaticity/antiaromaticity[5], collective spin excitations[6,7], enhanced nonlinear optical responses[8], and acting as molecular quantum rings[9]. Macrocycles with extended conjugation have attracted extensive attentions for their potential applications in organic solar cells, photodetectors, organic light-emitting diodes, drug discovery, and many others[2,10–12]. Moreover, owing to their structural rigidity and adaptivity, macrocycles can be assembled into one-dimensional tubular structures[13], lamellar assemblies[14], two-dimensional (2D) and three-dimensional (3D) organic crystals, which exhibit emerging properties such as semiconducting, porosity, etc.[14–16].

As an efficient method to fabricate organic nanostructures with atomic precision[17–19], on-surface synthesis has been used to synthesize poly-phenyl macrocycles in recent years[20–25]. The underlying premise of this bottom-up strategy is that precursor molecules undergo a well-defined sequence of inter- and intramolecular reactions, usually leading to the formation of one kind of macrocycle as the predominant product. In this work, we apply on-surface coupling reactions to synthesize a family of macrocycles using a precursor of tetraphenylethylene (TPE) derivative, 4,4′-(2,2-diphenylethene-1,1-diyl)bis(bromobenzene) (Br$_2$-TPE). TPE is a prototype aggregation-induced emission (AIE) chromophore[26,27]. Macrocycles made of TPE moieties linked with saturated bonds exhibit excellent fluorescence performance and improved selectivity and sensitivity in sensors due to the enhanced AIE effect[28–30]. Here, we synthesize the conjugated TPE-based macrocycles using a pseudo-high-dilution strategy[31]. As illustrated in Fig. 1, depositing the precursors onto an Ag(111) substrate held at an elevated temperature of 200 °C leads to the scission of C–Br bond[32] and formation of the macrocycles of different sizes (M4–M8) thanks to the conformational flexibility of TPE backbone. The yields of the even member macrocycles (M4, M6 and M8) are much higher than those of the odd member macrocycles, as revealed by the scanning tunneling microscopy (STM). Remarkably, three types of even-membered macrocycles (M4, M6 and M8) are spontaneously segregated from each other and assemble into mono-component supramolecular 2D crystals. The TPE macrocycles are closely packed in the 2D crystals, which potentially enhances the AIE effect. This work demonstrates that on-surface reaction can be employed to "one-pot" synthesis of multi-component macrocycles as the predominant products.

## Results

**Six-membered TPE macrocycles (M6).** Figure 2a shows a single domain of the 2D crystal made of M6 whose lateral size exceeds 200 nm. The crystalline structure displays six-fold symmetry, as evidenced with the fast Fourier transform (FFT) pattern shown in the inset. A magnified STM image shown in Fig. 2b reveals the 2D crystal is composed of closely packed hexagonal rings. The rhombic frame represents a unit cell ($a = b = 2.85 \pm 0.02$ nm, $\theta = 60 \pm 1°$), which is in good agreement with the FFT pattern (Supplementary Fig. 3). Some rings trap an X-shaped TPE molecule in their pores, as highlighted in the dashed circles. Figure 2c shows a hexagonal shaped M6 constituting six X-shaped TPE units that are positioned at the six corners of the hexagonal ring. Figure 2d is a schematic model of the M6 macrocycle (Note that the phenyl groups are rotated out of the plane in the actual structure). The edge of the hexagon (white arrow) is

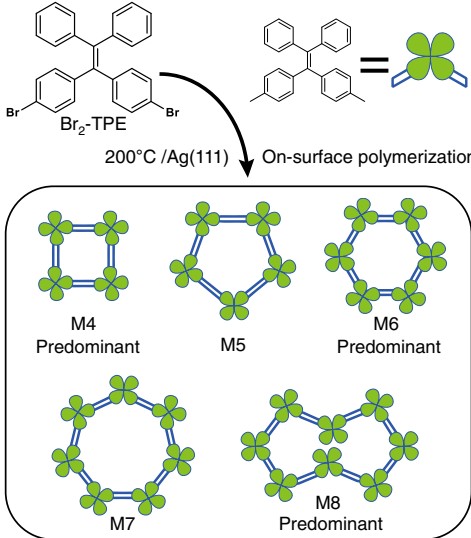

**Fig. 1 On-surface synthesis of tetraphenylethylene macrocycles.** Oligo-tetraphenylethylene macrocycles formed via the on-surface Ullmann coupling reaction of 4,4′-(2,2-diphenylethene-1,1-diyl)bis(bromobenzene) (Br$_2$-TPE) precursor.

0.98 ± 0.04 nm, which matches the model very well. The inner diameter of the hexagonal cavity (yellow arrow) is 1.69 ± 0.03 nm, which is large enough to host one TPE molecule. Figure 2e shows a schematic model of the unit cell. High-resolution STM image shown in Supplementary Fig. 2a confirms that no bromine atoms exist between M6 macrocycles, excluding the halogen-mediated self-assembly. Moreover, density-functional theory (DFT) optimized M6 conformer (Supplementary Fig. 2b) features tilted exterior phenyl groups. The exterior phenyl groups of the neighboring M6 macrocycles approach each other in a tilted T-shaped configuration with a ring-to-ring distance of 0.52 ± 0.02 nm (Supplementary Fig. 2a), which may invoke weak π–π interactions with the tilted T-shaped configurations[33,34]. Since each M6 macrocycle has 12 phenyl groups participating in the π–π interactions, presumably, the collective π–π interactions stabilize the 2D crystalline phase.

**Four-membered TPE macrocycles (M4).** Figure 3a shows a single domain of the 2D crystal made of M4 macrocycles, which also exceeds 200 nm. Both STM topograph and FFT pattern (shown in the inset) indicate this 2D crystal features a nearly square lattice. Several domain boundaries can be seen in Fig. 3a (see also Supplementary Fig. 5). Figure 3b is a magnified view of the area inside the square frame in Fig. 3a. A close inspection reveals that the unit cell of the 2D crystal is not a perfect square but slightly elongated (4%) in one direction: $a = 2.01 \pm 0.02$ nm, $b = 2.06 \pm 0.03$ nm, and $\theta = 90 \pm 1°$. The unit cells of the two domains are 90° rotated with respect to each other. Figure 3c shows a square shaped M4 macrocycle, which consists of four X-shaped TPE units at its four corners. Figure 3d is a schematic model of the M4 macrocycle. The edge length of ~1.0 nm matches the model. The cavity, however, features a rectangle instead of a square shape, indicating a less symmetric conformation. Figure 3e shows a schematic model of the unit cell. Similar to the M6 2D crystal, here the neighboring M4 macrocycles are subjected to π–π interactions between the tilted exterior phenyl groups of the neighboring M4 macrocycles (Supplementary Fig. 6a), with a ring-to-ring distance of 0.51 ± 0.04 nm.

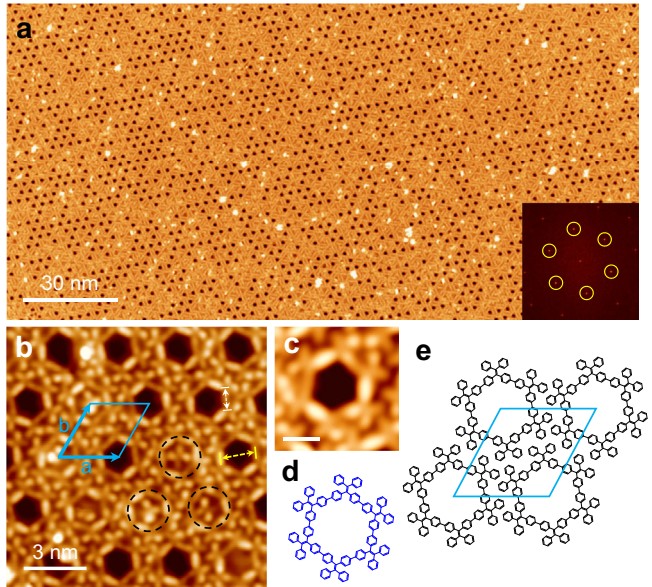

**Fig. 2 Six-membered TPE macrocycles (M6) formed on Ag(111). a** Large-scale STM image (200 nm × 100 nm; −1.5 V, 20 pA) of the 2D crystalline monolayer made of six-membered macrocycles (M6). Inset: FFT pattern. **b** Zoom in STM image (−1.0 V, 100 pA) of the M6 2D crystal. The blue rhomb denotes the unit cell. The edge (white arrow) and the inner diameter (yellow arrow) of the hexagon are 0.98 ± 0.04 nm and 1.69 ± 0.03 nm, respectively. Dashed black circles highlight inclusion of TPE monomers in the macrocycle cavities. **c**, **d** High-resolution STM image (0.2 V, 100 pA) and chemical model of M6. Scale bar: 1 nm. **e** Schematic model of the M6 2D crystal.

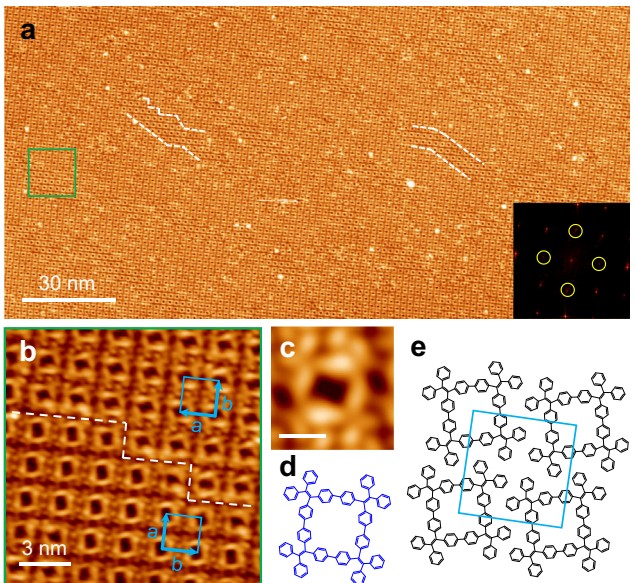

**Fig. 3 Four-membered TPE macrocycles (M4) formed on Ag(111). a** Large-scale STM image (200 nm × 100 nm; 2.0 V, 100 pA) of the 2D crystalline monolayer made of four-membered macrocycles (M4). Inset: FFT pattern. The dashed white lines indicate several domain boundaries. **b** Zoom in STM image (2.0 V, 200 pA) of the M4 2D crystal with a domain boundary (dashed line). The blue rectangles denote the unit cell of the 90° rotated domains. **c**, **d** High-resolution STM image (−1.0 V, 150 pA) and chemical model of M4. Scale bar: 1 nm. **e** Schematic model of the M4 2D crystal.

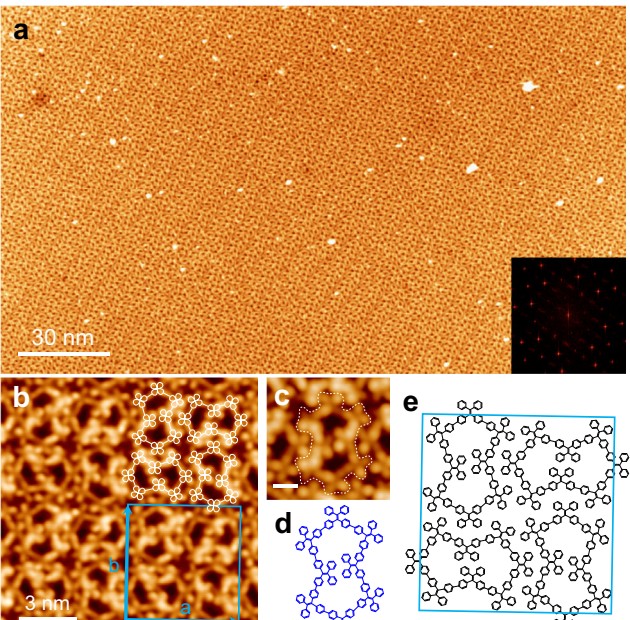

**Fig. 4 Eight-membered TPE macrocycles (M8) formed on Ag(111). a** Large-scale STM image (200 nm × 120 nm; −2.0 V, 20 pA) of the 2D crystalline monolayer made of eight-membered macrocycles (M8). Inset: FFT pattern. **b** Zoom in STM image (−1.0 V, 50 pA) of the M8 2D crystal with overlaid simplified molecular models. The blue square denotes the unit cell. **c**, **d** High-resolution STM image (−1.0 V, 150 pA) and chemical model of M8, outlined by the dashed white line. Scale bar: 1 nm. **e** Schematic model of the M8 2D crystal.

**Eight-membered TPE macrocycles (M8) with a Cassini oval shape**. Figure 4a shows a single domain of the 2D crystal made of M8 macrocycles. The FFT displays a very complicated pattern exhibiting four-fold symmetry. Figure 4b reveals that this structure is composed of Cassini oval-shaped M8 macrocycles. Different from the convex polygons of the smaller macrocycles of M4 or M6, M8 macrocycles are in a concave shape. The two voids of the M8 macrocycle have a comparable size as the cavity of the M4 macrocycle, both are too small to trap a TPE molecule. The concave shape significantly enlarges the packing density of M8. As illustrated with the overlaid M8 outline models, the neighboring M8 macrocycles are rotated by 90°, forming a close-packed basketweave pattern. Figure 4c shows the detail structural features of the M8 macrocycle. With the help of the schematic model shown in Fig. 4d, we can identify each X-shaped TPE unit in the M8 macrocycle. The two TPE units in the middle have their ethylene groups point inwardly, shaping a concave form. Moreover, the two middle TPE units are slightly offset, which breaks mirror symmetry. The adjacent M8 macrocycles along the diagonal directions in the basketweave pattern are mirror reflected. As a result, four M8 macrocycles form a basic unit of the 2D crystal. The blue square frame represents a unit cell ($a = b = 5.92 ± 0.08$ nm). Figure 4e presents a schematic model of a unit cell, showing the neighboring M8 macrocycles interact with each other via similar π–π interactions between the approaching exterior phenyl groups (ring-to-ring distance: 0.53 ± 0.04 nm).

Besides the three even-membered macrocycles, the on-surface coupling reaction also yields macrocycles consisting of five (M5) and seven (M7) TPE units, and open-end oligomer chains of different length. Supplementary Fig. 1 shows these structures are mixed on the substrate forming disordered monolayer. In comparison, the M6, M4 and M8 macrocycles are the

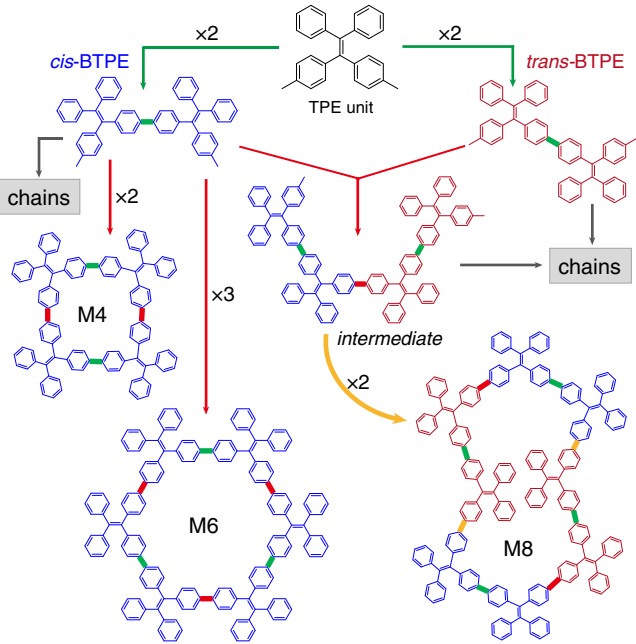

**Fig. 5 Proposed reaction pathways.** Multi-step reaction pathways for forming M4, M6, and M8 macrocycles.

predominant products and can assemble as the mono-component supramolecular 2D crystals with very large size. It is worth mentioning that the large-scale 2D crystalline islands made of M6, M4 and M8 macrocycles (shown in Figs. 2a, 3a, and 4a) are observed in different areas of the same sample. Therefore, the on-surface synthesized macrocycles are spontaneously segregated into mono-component domains. The M6 2D crystals can extend to micrometer size and cover nearly entire terraces, as shown in Supplementary Fig. 4. The 2D crystals fail to grow crossing the steps, indicating that only the width of the Ag(111) terrace limits the size of the 2D crystals. The M4 and M8 2D crystals are not as big as those of M6, nevertheless, can extend over ~20,000 nm², as shown in Figs. 3a and 4a. Note that we have not observed boundaries between the mono-component supramolecular islands even in the micrometer scale images.

**Multi-step reaction pathways**. Here we discuss the possible reaction pathways of M4, M6 and M8. As illustrated in Fig. 5, we propose a multi-step ring formation mechanism: the first step is coupling of two TPE monomers forming a dimer (BTPE), as represented with the green links. The dimer can be a *cis*-conformation or a *trans*-conformation. Next, coupling of two (three) *cis*-BTPEs, as represented with the red links, results in an M4 (M6) ring. *cis*-BTPEs can be coupled to form open-end oligomer chains too. Coupling of *trans*-BTPEs always forms open-end oligomer chains. Interestingly, cross-coupling of a *cis*-BTPE and a *trans*-BTPE forms an intermediate, further coupling of two intermediates, as represented with the orange links, may yield an M8 macrocycle with a Cassini oval shape. The products of M5 or M7 can be formed by ring-closing coupling of one TPE unit with two or three *cis*-BTPEs (Supplementary Fig. 7), respectively.

To date, it has been reported that on-surface synthesis always yields only one predominant macrocycle product[20,21,23]. For example, a precursor with *meta*-linkage yields hexagonal macrocycles. We attribute the multiple products in our work to the structural flexibility of the TPE backbone thanks to its non-planar conformation. The DFT optimized structure of a Br₂-TPE precursor reveals that the opening angle between the two

bromobenzene groups (denoted as ph-ethy-ph angle) is ~114.65°, as shown in Supplementary Fig. 8a, which favors hexagonal shaped M6. In contrast, the M4 and M8 macrocycles contain the TPE units with ph-ethy-ph angles deviated from 120° owing to the deformed TPE backbones. Note that recent optical studies demonstrate that the emission behaviors of TPE units vary against conformation changes[35,36]. Therefore, the M4, M6, and M8 macrocycles, which consist of TPE units with different conformations, provide a platform to explore conformation-dependent photophysical properties.

## Discussion

The growth of such large-area mono-component 2D supramolecular crystals is very rare in a multi-component system. We propose that in these mono-component 2D crystals the periphery phenyl groups of neighboring macrocycles are maximally interconnected, specifically, the hexagonal M6 in the three-fold packing, the square M4 in the four-fold packing, and the Cassini oval shape M8 in the basketweave packing, so that the mono-component 2D crystals are energetically favored. The assembly process, which takes place at 200 °C, is highly dynamic, involving spontaneous segregation and self-assembly of the as-formed macrocycles on the surface. The segregation requires mutual recognition of different-type macrocycles, while the self-assembly requires self-recognition of same-type macrocycles. Without direct experimental observation or sophisticated theoretical simulation, this intricate process is beyond our comprehension. We propose two possible growth modes: (1) the macrocycles are randomly mixed at 200 °C and undergo phase separation in the cooling process, resulting in the mono-component supramolecular islands; (2) segregation and self-assembly of the macrocycles occur at 200 °C. Since we do not have direct experimental data revealing the high-temperature process, we cannot draw a conclusion which one is true. Nevertheless, the huge size of the mono-component islands and absence of domain boundaries imply that the segregation and self-assembly are high effective, so we incline to the latter scenario.

In summary, we demonstrate "one-pot" synthesis of 4-, 6-, and 8-membered TPE macrocycles on a Ag(111) surface as the predominant products, benefiting from the structural flexibility of TPE backbone. These macrocycles are spontaneously segregated on the surface and self-assemble as mono-component 2D supramolecular crystals with very large areas. Our findings provide a design strategy toward precise synthesis of multi-component macrocycles on a surface. We foresee that the large-area synthesis of shape-persistent conjugated oligo-TPE macrocycles will enable exploration of their photophysical properties using STM-induced luminescence[37–39].

## Methods

**Sample preparation and STM measurements**. All experiments were carried out in an ultrahigh vacuum system (base pressure <3.0 × 10⁻¹⁰ mbar) equipped with a CreaTec low-temperature STM. A single-crystalline Ag(111) substrate was cleaned via repeated cycles of Ar⁺ sputtering (0.8 keV) and subsequent annealing at 450 °C. Molecules of 4,4′-(2,2-diphenylethene-1,1-diyl)bis(bromobenzene) were sublimated from a Knudsen cell at 140 °C, to achieve a deposition rate of ~0.66 ML/h, while the substrate was kept at 200 °C during molecule deposition. A monolayer is defined as the amount of deposited molecules that entirely covers the substrate surface. The slow deposition on a hot surface was used to promote the yield of macrocycles by achieving pseudo-high-dilution[31]. After growth, the sample was transferred to STM chambers for STM measurements. The STM images were taken at either 77 K or 5.3 K in the constant-current mode. All the lateral distances are measured from the STM images at 5.3 K.

**Theoretical calculations**. All DFT calculations were performed by using Gaussian 09 program[40] using the B3LYP method with the 6–31G(d,p) basis set for structure optimization. Molecular geometries are generated by IQmol molecular viewer.

## Data availability
The datasets generated during and/or analyzed during the current study are available from the corresponding author on reasonable request.

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

## Acknowledgements
This work is financially supported by the Research Grants Council of Hong Kong (C6014-20W and 16301219) and the Innovation and Technology Commission (ITC-CNERC14SC01).

## Author contributions
N.L. conceived the project; E.L., C.K.L., and C.C. performed the on-surface synthesis and STM experiments; H.X. synthesized the precursor molecules under the supervision of J.W.Y.L. and B.Z.T.; E.L. performed DFT calculations with assistance from J.Z.; the manuscript was written by E.L. and N.L. with contributions from all co-authors.

## Competing interests
The authors declare no competing interests.
