## [Peer Review File · Communications Chemistry]

Reviewers' comments:

Reviewer #1 (Remarks to the Author):

In this manuscript, Li et al demonstrate a "one-pot" method towards high-yield synthesis and separation of a variety of macrocycles (M4, M6 and M8) from a conformational flexible molecular precursor Br₂-TPE on Ag(111) by using STM measurements. There exist high-yields of even numbered macrocycles, while odd-number ones (M5 and M7) have much lower yields. Interestingly, the authors show a mono-component assembly of the even-number macrocycles, and tentatively attribute this structural separation to conformational flexibility and the pi-pi interactions between peripheral phenyls. The STM data are well presented, the results and discussion support the paper's major conclusions. We recommend the publication after the authors addressing following revisions:

1. The authors named the as-formed macrocycles "giant conjugated" in title and abstract section, and did not mention it any more in main text. I suggest not using "giant".
2. I did not find out Br atoms in any STM images. Where are Br atoms after the on-surface reaction? On Ag(111), there also existed C-Ag organometallic intermediates in previous literatures. Did the authors observe the intermediates?
3. About the proposed multi-step ring formation mechanism in Scheme 2, page 8, I suggest more discussion. This comment is also related to Comment 2. Do Br atoms interact with macrocycles, or assist their separation? The author may give literatures, or experimental results to support the proposed pi-pi interaction assisting formation of mono-component assembly, which repeatedly appeared after each discussion of even numbered macrocycles. Do C-Ag intermediates, if exist, contribute to the formation and separation of macrocycles?
4. In discussion on Figure 1d and e, some length measurements are not well presented, please indicate the lengths/distances in figures.
5. The author used pseudo-high-dilution approach (Ref. 35) to promote yields of macrocycles. I suggest the authors citing the reference at the end of the introduction section.
6. In line 5, page 9, the authors may provide references or theoretical calculations on "The DFT optimized structure of a Br₂-TPE...". I also suggest to provide STM data on intact molecular precursors in SI.
7. In sample preparation section, the author should define one monolayer (ML). In last line, page 3, "Van-der Waals" should be "van der Waals".

Reviewer #2 (Remarks to the Author):

The manuscript "On-surface synthesis and separation of giant conjugated tetraphenylethylene macrocycles", by Li et al., details an on-surface synthesis protocol for the formation of large conjugated macrocycles, from halogen functionalised tetraphenylethylene monomers, which are observed to spontaneously separate into homogenous mono-component domains on the Ag(111) surface. The molecule-substrate system is characterised by ultra-high vacuum scanning tunnelling microscopy (STM) undertaken at low temperature (77K, 5.3 K); which is used as the primary method for assigning the structure of the macrocycles and the packing of the extended self-assembled molecular domains. The exact chemical structure of the on-surfaced synthesised macrocycles, and the conformation of the reaction products, are not unambiguously demonstrated, but the models are self-consistent with respect to the STM data acquired and the good agreement with the expected reaction products for Ullmann coupling of the monomers strongly support the conclusions of the manuscript.

The work presented in the manuscript is novel and timely. On-surface synthesis is a field with broad

appeal and the spontaneous ordering of the reaction products described in the work, giving rise to large areas of ordered material, is a significant result.

I recommended that the manuscript be published, following consideration of the following points:

- 1) On page 3 the authors state that the dimensions of the M6 supramolecular structure obtained from the STM images are in good agreement with the FFT data. The FFT obtained values should be cited with additional information (if relevant) in the supplementary materials.
- 2) The authors discuss the trapping of TPE within the macrocycle pores (shown in Figure 1). Can the authors show STM data for the unreacted and/or debrominated TPE species on the Ag(111) surface?
- 3) On page 4 there is a discussion of the role of π - π interactions in stabilising the self-assembled structures. The authors should provide additional information (and/or references) to support the proposed titling of the phenyl groups.
- 4) Figure 2 is stated to show domain boundaries – examples of these should be indicated in the figure (and an example of the two different unit cells could be included).
- 5) Can the authors comment on the boundaries between ordered M8, M4, and M6 regions? Are these boundaries frequently observed in the STM data?
- 6) Do the authors observe the linear oligomers being formed as a stable product within the experiment? If so data of this should be provided within the supplementary materials.
- 7) DFT calculations of the optimised structures are discussed – details of these should be provided within the supplementary materials.
- 8) With regards to the self-assembly of mono-component structures, can the authors clarify under what conditions the separation occurs? At 200deg C do the authors assume that macrocycles may intermix freely and then separation occurs during cooling, or that the separation has already occurred at 200 deg C? Can the authors comment on how the cooling rate of the sample may affect the quality/type of self-assembled structures formed?
- 9) Related to the above point. Do the authors note any difference in the self-assembled structures between samples imaged at 77 K or 5.3 K? The method section suggests that samples were imaged at both of these temperatures. Any differences should be commented upon in the manuscript.
- 10) The authors motivate their work in terms of the properties and applications of macrocycles with extended conjugation. The following manuscripts could also be cited within this context - <https://doi/10.1021/acs.accounts.8b00313> <https://doi.org/10.1103/PhysRevLett.125.206803>.

Reviewer #3 (Remarks to the Author):

The manuscript by Li and coworkers reports their observations on the surface-thermally induced cyclization of a simple dibromophenylethene derivative. The manuscript shows some attractive STM images and highlights the segregation of the products resulting from the on-surface synthesis. However, the authors should consider the following points:

- 1) The manuscript has the feeling of being just a list of observations and there us very little actual analysis of the structures and the connotations of their formation or properties. No DFT optimized structures are presented in support.
- 2) At several points, the authors speculate about the possible significance of the macrocycles for AIE but no relevant data is presented. It is OK to mention relevance of AIE to TPE compounds in the introduction but these claims for the presented compounds should be supported by actual data of the isolated compounds.
- 3) The authors have used the term 'separation' to describe the crystallization of the even-numbered macrocycles in different domains. I object strongly to the use of this term since the authors have not separated the compounds in any actually physically meaningful way.

4) A mechanism is presented for the formation of even numbered macrocycles, which is easy to comprehend based on the molecular geometry. What about the much more interesting case of the odd-numbered macrocycles? How are they formed?

5) The authors mention 'yield' and frequently 'high-yield'. However, no estimation of yields or relative yield is given and I found no evidence for this claim based on any of the STM image data.

6) The authors support their purported high yields by invoking DFT optimized structures which are not shown. I suspect that they have referred to previous work and have neglected to add the citation. However, to establish the reactivity patterns on Ag(111) will require analysis of DFT optimized structures of the compound on that surface. This is essential to help validate the several so far unsubstantiated claims that the authors have made in this case.

7) The authors seem keen to emphasize the growth of segregated mono-component domains of the on-surface chemistry products calling it 'phenomenal'. However, I was not especially surprised by this feature since co-deposition of mixtures of molecules often ends at segregated states. One interesting feature of those systems is what occurs at the phase boundaries. Unfortunately, this is not addressed here and only the mixed phase of M5/M7 is shown. Given the chemical similarities between the macrocycles, it could be a point of interest.

Overall, given the lack of in-depth analysis of these quite interesting structures, I cannot suggest acceptance of this work in its current state.

Point-by-point response to reviewers' comments

The responses to the reviewers' comments are highlighted in blue and the changes in the revised manuscript are highlighted in red.

Response to Reviewer 1#

Comment: In this manuscript, Li et al demonstrate a “one-pot” method towards high-yield synthesis and separation of a variety of macrocycles (M4, M6 and M8) from a conformational flexible molecular precursor Br₂-TPE on Ag(111) by using STM measurements. There exist high-yields of even numbered macrocycles, while odd-number ones (M5 and M7) have much lower yields. Interestingly, the authors show a mono-component assembly of the even-number macrocycles, and tentatively attribute this structural separation to conformational flexibility and the pi-pi interactions between peripheral phenyls. The STM data are well presented, the results and discussion support the paper's major conclusions. The We recommend the publication after the authors addressing following revisions:

Response: We thank the reviewer for his/her positive evaluation of our work.

1. The authors named the as-formed macrocycles “giant conjugated” in title and abstract section, and did not mention it any more in main text. I suggest not using “giant”.

Response: We thank the suggestion of the referee and have deleted it in the revised manuscript.

Title:

On-surface synthesis and spontaneously segregation of ~~giant~~ conjugated tetraphenylethylene macrocycles

Abstrate:

“Creating conjugated ~~giant~~ macrocycles has attracted extensive interest because the unique chemical...”

2. I did not find out Br atoms in any STM images. Where are Br atoms after the on-surface reaction? On Ag(111), there also existed C-Ag organometallic intermediates in previous literatures. Did the authors observe the intermediates?

Response: As shown in Fig. S2a in revised SI, we do observe some dot-like features inside the cavities, which may be Br atoms. We do not observe any features that represent Br atoms existing between the M6 macrocycle. Desorption of Br atoms from the Ag(111) surface occurs at 200°C (JACS, **141**, 4824-4832 (2019)).

Figure S2: (a) High-resolution STM image (-0.55V , 50 pA) of an M6 island. The protrusions inside the cavities, indicated by the white arrows, are attributed to Br adatoms. The measured distance between neighboring exterior phenyl groups, indicated by the dashed red lines, is $0.52\pm 0.02\text{ nm}$. (b) Structure of a planar M6 conformer optimized by DFT method at B3LYP/6-31G(d,p) level, Gaussian 09 program. The side-to-side length of the hexagon is 1.73 nm , agreeing well with the measured value ($1.69\pm 0.03\text{ nm}$).

C-Ag organometallic intermediates usually emerge at lower annealing temperature prior to covalent coupling and are converted to C-C products at 150°C (JACS, **141**, 4824-4832 (2019)). Figure R1 shows the STM image after precursors deposition at RT followed by annealing at 100°C, displaying the dimers, trimers, and unreacted monomers. The dashed white line marks a possible C-Ag organometallic intermediate.

Figure R1: STM image (-1.0V , 50 pA) after deposition of $\text{Br}_2\text{-TPE}$ precursor on $\text{Ag}(111)$ at room temperature and annealing the sample at 100°C .

3. About the proposed multi-step ring formation mechanism in Scheme 2, page 8, I suggest more discussion. This comment is also related to Comment 2. Do Br atoms interact with macrocycles, or assist their separation? The author may give literatures, or experimental results to support the proposed $\pi\text{-}\pi$ interaction assisting formation of mono-component assembly, which repeatedly appeared after each discussion of even numbered macrocycles. Do C-Ag intermediates, if exist, contribute to the formation and separation of macrocycles?

Response: We thank the suggestion of the referee. High-resolution STM image shown in Fig. S2a (revised SI) reveal that Br atoms do not exist between the M6 macrocycles, excluding the halogen-mediated self-assembly. Moreover, we have also added the DFT-optimized M6 conformer, as shown in Fig. S2b. The exterior phenyl groups of M6 are rotated out of the plane in the optimized structure. The measured distance between neighboring exterior phenyl groups is $0.52\pm 0.02\text{ nm}$ for M6, $0.51\pm 0.04\text{ nm}$ for M4, and $0.53\pm 0.04\text{ nm}$ for M8 2D crystal. The values agree well with the reported center-to-center distance of benzene dimer with T-shaped configurations ($4.9\text{-}5.1\text{ \AA}$) or tilted T-shaped configurations ($4.7\text{-}4.97\text{ \AA}$) [JACS, **124**, 10887-10893 (2002); J. Phys. Chem. A, **111**, 3446-3457 (2007)].

C-Ag intermediates do not exist in the self-assembly process (see detailed discussion below), thus do not contribute to the formation and separation of macrocycles.

In the revised manuscript, we have added the following discussions:

“Figure 1e shows a schematic model of the unit cell. High-resolution STM image shown in Figure S2a confirms that no bromine adatom exist between M6 macrocycles, excluding the halogen-mediated self-assembly. Moreover, density-functional theory (DFT) optimized M6 conformer (Figure S2b) features tilted exterior phenyl groups. The exterior phenyl groups of the neighboring M6 macrocycles approach each other in a tilted T-shaped configurations with a ring-to-ring distance of $0.52\pm 0.02\text{ nm}$ (Figure S2a), which may invoke weak $\pi\text{-}\pi$ interactions with the tilted T-shaped configuration^{33, 34}. Since each M6 macrocycle has 12 phenyl groups participating in the $\pi\text{-}\pi$ interactions,…”

[33] Sinnokrot MO, Valeev EF, Sherrill CD. Estimates of the Ab Initio Limit for π - π Interactions: The Benzene Dimer. *J. Am. Chem. Soc.* 124, 10887-10893 (2002).

[34] Lee EC, Kim D, Jurečka P, Tarakeshwar P, Hobza P, Kim KS. Understanding of Assembly Phenomena by Aromatic-Aromatic Interactions: Benzene Dimer and the Substituted Systems. *J. Phys. Chem. A* 111, 3446-3457 (2007).

“...Similar to the M6 2D crystal, here the neighboring M4 macrocycles are subjected to π - π interactions between the tilted exterior phenyl groups of the neighboring M4 macrocycles (Figure S6a), with a ring-to-ring distance of 0.51 ± 0.04 nm.”

“...,showing the neighboring M8 macrocycles interact with each other via similar π - π interactions between the approaching exterior phenyl groups (ring-to-ring distance: 0.53 ± 0.04 nm).”

“...The products of M5 or M7 can be formed by ring-closing coupling of one TPE unit with two or three *cis*-BTPEs (Scheme S1), respectively. The low yield of odd-numbered macrocycles may be attributed to the high yield of *cis*-BTPE under pseudo-high-dilution condition before encountering a third precursor molecule.”

4. In discussion on Figure 1d and e, some length measurements are not well presented, please indicate the lengths/distances in figures.

Response: We have taken the reviewer’s suggestions and added labels in the revised Figure 1b and Figure S2:

Figure 1. (b) Zoom in STM image (-1.0 V, 100 pA) of the M6 2D crystal. The blue rhomb denotes the unit cell. The edge (white arrow) and the inner diameter (yellow arrow) of the hexagon are 0.98 ± 0.04 nm and 1.69 ± 0.03 nm, respectively. Dashed black circles highlight inclusion of TPE

monomers in the macrocycle cavities.

5. The author used pseudo-high-dilution approach (Ref. 35) to promote yields of macrocycles. I suggest the authors citing the reference at the end of the introduction section.

Response: We thank the suggestion of the referee and have cited it in the revised introduction section: “...Here, we synthesize the conjugated TPE-based macrocycles using a pseudo-high-dilution strategy³⁰.”

6. In line 5, page 9, the authors may provide references or theoretical calculations on “The DFT optimized structure of a Br₂-TPE...”. I also suggest to provide STM data on intact molecular precursors in SI.

Response: We thank the suggestion of the referee. In the revised manuscript, we have added both the DFT optimized structure of Br₂-TPE and the STM image:

Figure S4: (a) DFT optimized geometry of Br₂-TPE, displaying a propeller-like structure. The angle between two phenyl rings connected with the central ethylene, as marked by a red arc arrow, is ~114.65°. (b) STM image (-1.5V, 51 pA) of Br₂-TPE precursors deposited onto Ag(111) held at room temperature, which produces disordered islands. Structural models are superimposed on the image.

7. In sample preparation section, the author should define one monolayer (ML). In last line, page 3, “Van-der Waals” should be “van der Waals”.

Response: We have taken the reviewer’s suggestions and added more description as follows:

“...deposition. A monolayer is defined as the amount of deposited molecules that entirely covers the substrate surface. The slow deposition...”

We thank the reviewer’s careful checks and have fixed the typo about “van der Waals”.

“...matches the model very well. The inner diameter of the hexagonal cavity is 1.69±0.03 nm...”

Response to Reviewer 2#

The manuscript “On-surface synthesis and separation of giant conjugated tetraphenylethylene macrocycles”, by Li et al., details an on-surface synthesis protocol for the formation of large conjugated macrocycles, from halogen functionalised tetraphenylethylene monomers, which are observed to spontaneously separate into homogenous mono-component domains on the Ag(111) surface. The molecule-substrate system is characterised by ultra-high vacuum scanning tunnelling microscopy (STM) undertaken at low temperature (77K, 5.3 K); which is used as the primary method for assigning the structure of the macrocycles and the packing of the extended self-assembled molecular domains. The exact chemical structure of the on-surfaced synthesised macrocycles, and the conformation of the reaction products, are not unambiguously demonstrated, but the models are self-consistent with respect to the STM data acquired and the good agreement with the expected reaction products for Ullmann coupling of the monomers strongly support the conclusions of the manuscript.

The work presented in the manuscript is novel and timely. On-surface synthesis is a field with broad appeal and the spontaneous ordering of the reaction products described in the work, giving rise to large areas of ordered material, is a significant result.

Response: We thank the reviewer for his/her positive evaluation of our work.

I recommended that the manuscript be published, following consideration of the following points:

1) On page 3 the authors state that the dimensions of the M6 supramolecular structure obtained from the STM images are in good agreement with the FFT data. The FFT obtained values should be cited with additional information (if relevant) in the supplementary materials.

Response: We have taken the reviewer’s suggestions and added the inverse FFT image/values in the revised supplementary materials:

Figure S3: (a) STM topography (-1.8V , 10 pA) of an M6 island. (b) Fast Fourier transform (FFT) image of (a). (c) Real space image after filtering and inverse FFT based on the circled Bragg peaks in the FFT shown in (b). The blue rhombic frames in (a) and (c) mark the same position, denoting the unit cell. (d) Line profiles along the black line in (c), showing a measured period of 2.84 nm .

2) The authors discuss the trapping of TPE within the macrocycle pores (shown in Figure 1). Can the authors show STM data for the unreacted and/or debrominated TPE species on the Ag(111) surface?

Response: We thank the suggestion of the referee. In the revised manuscript, we have added STM data of unreacted precursors after deposition onto Ag(111) at room temperature:

Figure S4: (a) DFT optimized geometry of Br₂-TPE, displaying a propeller-like structure. The angle between two phenyl rings connected with the central ethylene, as marked by a red arc arrow, is ~114.65°. (b) STM image (-1.5V, 51 pA) of Br₂-TPE precursors deposited onto Ag(111) held at room temperature, which produces disordered islands. Structural models are superimposed on the image.

3) On page 4 there is a discussion of the role of π - π interactions in stabilising the self-assembled structures. The authors should provide additional information (and/or references) to support the proposed titling of the phenyl groups.

Response: We thank the suggestion of the referee. We have added the DFT optimized M6/M4/M8 conformers in the revised supplementary materials. As shown in Figure S2b and Figure S6, the exterior phenyl groups of all these macrocycles are rotated out of the plane in the optimized structure. Furthermore, the measured distance between neighboring exterior phenyl groups is 0.52 ± 0.02 nm for M6, 0.51 ± 0.04 nm for M4, and 0.53 ± 0.04 nm for M8 2D crystal. The values agree well with the reported center-to-center distance of benzene dimer with T-shaped configurations (4.9 - 5.1 Å) or tilted T-shaped configurations (4.7 - 4.97 Å) [JACS, **124**, 10887-10893 (2002); J. Phys. Chem. A, **111**, 3446-3457 (2007)].

Figure S2: (a) High-resolution STM image (-0.55V, 50 pA) of an M6 island. The protrusions inside the cavities, indicated by the white arrows, are attributed to Br atoms. The measured distance between neighboring exterior phenyl groups, indicated by the dashed red lines, is 0.52 ± 0.02 nm. (b) Structure of a planar M6 conformer optimized by DFT method at B3LYP/6-31G(d,p) level, Gaussian 09 program. The side-to-side length of the hexagon is 1.73nm, agreeing well with the measured value (1.69 ± 0.03 nm).

Figure S6: (a) Top and side views of a DFT optimized M4 conformer, displaying a planar configuration with tilted exterior phenyl groups. (b) Top and side views of a DFT optimized M8 conformer. Two inward TPE units are spatially separated, resulting in a nonplanar configuration.

4) Figure 2 is stated to show domain boundaries – examples of these should be indicated in the figure (and an example of the two different unit cells could be included).

Response: We have taken the reviewer's suggestions and added the domain boundaries in the revised Figure 2a. Moreover, we have also added the Figure S5, showing the rotated domains clearly.

Figure 2. Four-membered TPE macrocycles (M4) formed on Ag(111). (a) Large-scale STM image (200nm×100nm; 2.0 V, 100 pA) of the 2D crystalline monolayer made of four-membered macrocycles (M4). Inset: FFT pattern. The dashed white lines indicate several domain boundaries.

Figure S5: (a) Large-scale STM image ($V=-0.9$ V, $I=50$ pA) of an M4 island. Inset: FFT pattern. (b) Zoom in STM image of the M4 2D crystal with a domain boundary (dashed line). The blue rectangles denote the unit cell of the 90° rotated domains.

5) *Can the authors comment on the boundaries between ordered M8, M4, and M6 regions? Are these boundaries frequently observed in the STM data?*

Response: The large-scale STM images show that the macrocycles always assemble into large-area mono-component supramolecular islands. In particular, the M6 2D crystals can extend to micrometer size and cover nearly entire terraces (Figure S4). We have not observed boundaries between the mono-component supramolecular islands even in the micrometer scale images. This is different from the common cases for on-surface synthesis: evenly distributed nanoscale islands [e.g., ACS Nano 12, 12612-12618 (2018)]. The large-area single domains observed in our work implies a unique growth mode (see detailed discussion in answer for Q8.)

6) *Do the authors observe the linear oligomers being formed as a stable product within the experiment? If so data of this should be provided within the supplementary materials.*

Response: Besides the large-size ordered M6, M4, and M8 regions, the oligo-TPE chains and various macrocycles are mixed together as the disordered islands. Figure S1e shows such a disordered mixture of dimer, trimer, M10, and complex linear oligomers:

Figure S1: (e) Right panel: STM image (-1.15 V, 100 pA) showing disordered mixture of oligo-TPE chains, and odd-number macrocycles. The dashed rectangle marks a Cassini oval-shaped M10 macrocycle. **Left panel: corresponding chemical models of dimer, trimer, and M10.**

7) *DFT calculations of the optimised structures are discussed – details of these should be provided within the supplementary materials.*

Response: We thank the suggestion of the referee. We have added the DFT optimized M6/M4/M8 conformer in the revised supplementary materials: Figure S2b for M6, Figure S6 for M4/M8, and Figure S7 for Br₂-TPE.

8) *With regards to the self-assembly of mono-component structures, can the authors clarify under what conditions the separation occurs? At 200deg C do the authors assume that macrocycles may intermix freely and then separation occurs during cooling, or that the separation has already occurred at 200 deg C? Can the authors comment on how the cooling rate of the sample may affect the quality/type of self-assembled structures formed?*

Response: We propose two possible growth modes: (1) the macrocycles are randomly mixed at 200°C and undergo phase separation in the cooling process, resulting in the mono-component supramolecular islands; (2) segregation and self-assembly of the macrocycles occur at 200°C. Since we do not have direct experimental data revealing the high-temperature process, we cannot draw a conclusion which one is true. Nevertheless, the huge size of the mono-component islands and absence of domain boundaries imply that the segregation and self-assembly are high effective, which makes us incline to the latter scenario. In this case, cooling rate does not affect the final

structures.

In revised manuscript, we have revised the discussion as follows:

“The assembly process, which takes place at 200°C, is highly dynamic, involving spontaneous segregation and self-assembly of the as-formed macrocycles on the surface. The segregation requires mutual recognition of different-type macrocycles, while the self-assembly requires self-recognition of same-type macrocycles. Without direct experimental observation or sophisticated theoretical simulation, this intricate process is beyond our comprehension. We propose two possible growth modes: (1) the macrocycles are randomly mixed at 200°C and undergo phase separation in the cooling process, resulting in the mono-component supramolecular islands; (2) segregation and self-assembly of the macrocycles occur at 200°C. Since we do not have direct experimental data revealing the high-temperature process, we cannot draw a conclusion which one is true. Nevertheless, the huge size of the mono-component islands and absence of domain boundaries imply that the segregation and self-assembly are high effective, so we incline to the latter scenario.”

9) *Related to the above point. Do the authors note any difference in the self-assembled structures between samples imaged at 77 K or 5.3 K? The method section suggests that samples were imaged at both of these temperatures. Any differences should be commented upon in the manuscript.*

Response: In our experiments, we wait the sample cool down to room temperature after growth. Then the sample was transferred to STM low-temperature stage (77K or 5.3K) for imaging. There is no structural difference for the sample imaged at 77K or 5.3K. All the lateral distances in this manuscript are measured from the STM images acquired at 5.3K, to ensure the accuracy.

In revised manuscript, we have added the related descriptions as follows:

“...The STM images were taken at either 77 K or 5.3 K in the constant-current mode. All the lateral distances are measured from the STM images at 5.3K.”

10) *The authors motivate their work in terms of the properties and applications of macrocycles with extended conjugation. The following manuscripts could also be cited within this context - <https://doi/10.1021/acs.accounts.8b00313> <https://doi.org/10.1103/PhysRevLett.125.206803>.*

Response: We have taken the reviewer’s suggestions and added the two interesting papers in the

revised manuscript:

“...inner cavities of shape-persistent macrocycles can host molecules or ions forming host-guest supramolecular complexes^{3,4}. In contrast to the open-chain oligomers, the cyclic topology gives rise to unique electronic and optical properties such as aromaticity/antiaromaticity⁵, collective spin excitations^{6,7}, enhanced nonlinear optical responses⁸, and acting as molecular quantum rings⁹...”

[4] Bols PS, Anderson HL. Template-Directed Synthesis of Molecular Nanorings and Cages. *Acc. Chem. Res.* **51**, 2083-2092 (2018).

[9] Judd CJ, et al. Molecular Quantum Rings Formed from a π -Conjugated Macrocyclic. *Phys. Rev. Lett.* **125**, 206803 (2020).

Response to Reviewer 3#

The manuscript by Li and coworkers reports their observations on the surface-thermally induced cyclization of a simple dibromophenylethene derivative. The manuscript shows some attractive STM images and highlights the segregation of the products resulting from the on-surface synthesis.

Response: We thank the reviewer for the positive remarks on the significance and quality of our work.

However, the authors should consider the following points:

1) The manuscript has the feeling of being just a list of observations and there us very little actual analysis of the structures and the connotations of their formation or properties. No DFT optimized structures are presented in support.

Response: So far, to the best of our knowledge, all the reported on-surface synthesis always yields only one kind of macrocycle as the predominant product. Our work reports the “one-pot” synthesis of three kinds of TPE macrocycles that self-assemble as large-area mono-component supramolecular islands. We believe these findings represent a quite novel phenomenon in on-surface synthesis. We also thoroughly analyze the structures of the individual macrocycles as well as the mono-component supramolecular islands using STM and DFT.

Following the suggestion of the referee, we have added the DFT optimized M6/M4/M8 conformer in the revised supplementary materials: Figure S2b for M6, Figure S6 for M4/M8, and Figure S7 for Br₂-TPE.

Figure S2: (b) Structure of a planar M6 conformer optimized by DFT method at B3LYP/6-31G(d,p) level, Gaussian 09 program. The side-to-side length of the hexagon is 1.73nm, agreeing well with the measured value (1.69 ± 0.03 nm).

Figure S6: (a) Top and side views of a DFT optimized M4 conformer, displaying a planar configuration with tilted exterior phenyl groups. (b) Top and side views of a DFT optimized M8 conformer. Two inward TPE units are spatially separated, resulting in a nonplanar configuration.

Figure S7: (a) DFT optimized geometry of Br₂-TPE, displaying a propeller-like structure. The angle between two phenyl rings connected with the central ethylene, as marked by a red arc arrow, is ~114.65°.

2) At several points, the authors speculate about the possible significance of the macrocycles for AIE but no relevant data is presented. It is OK to mention relevance of AIE to TPE compounds in the introduction but these claims for the presented compounds should be supported by actual data of the isolated compounds.

Response: We agree with the referee that it is highly desirable to explore the photophysical

properties of the synthesized oligo-TPE macrocycles (such as conformation-dependent; isolated or self-assembled). We plan to work along this direction by using STM-induced luminescence in future. Considering the current manuscript mainly focuses on the synthesis, we believe it is appropriate to present our work without including optical properties.

3) The authors have used the term 'separation' to describe the crystallization of the even-numbered macrocycles in different domains. I object strongly to the use of this term since the authors have not separated the compounds in any actually physically meaningful way.

Response: We thank the suggestion of the referee and have revised the related descriptions as follows:

Title:

“On-surface synthesis and spontaneously **segregation** of conjugated tetraphenylethylene macrocycles”

Abstract part:

“...on-surface reaction. The as-synthesized macrocycles are spontaneously **segregated** on the surface and self-assemble as two-dimensional mono-component supramolecular...”

Introduction part:

“Remarkably, the three even-membered macrocycles of different sizes spontaneously **segregated** from each other and assemble into mono-component supramolecular 2D crystals.”

Main part:

“...the same sample. Therefore, the on-surface synthesized macrocycles are spontaneously **segregated** into homo-component domains. Besides the three even-membered...”

“...assembly process, which takes place at 200°C, is highly dynamic. This process requires highly effective spontaneous **segregation** and self-assembly of the as-formed...”

“...flexibility of TPE backbone. These macrocycles are spontaneously **segregated** on the surface and self-assemble as mono-component 2D supramolecular crystals with very large areas.”

4) A mechanism is presented for the formation of even numbered macrocycles, which is easy to comprehend based on the molecular geometry. What about the much more interesting case of the

odd-numbered macrocycles? How are they formed?

Response: As shown in Scheme S1, the odd-numbered macrocycles, like M5 or M7, can be formed by ring-closing coupling of one TPE unit with two or three cis-BTPEs, respectively.

Scheme S1. Proposed reaction pathways for forming M5 and M7 macrocycles.

5) The authors mention 'yield' and frequently 'high-yield'. However, no estimation of yields or relative yield is given and I found no evidence for this claim based on any of the STM image data.

Response: We thank the referee for this comment. As the large-scale STM images shown in main text, the as-formed TPE macrocycles always assemble into large-area islands. In particular, the M6 2D crystals can extend to micrometer size and cover nearly entire terraces (Figure S4). The large-area growth observed in our work makes it difficult to quantify the yield/relative yield of macrocycles, due to the size limitation of STM imaging. However, our STM data clearly confirm that the yields of the even member macrocycles (M4, M6 and M8) are much higher than those of the odd member macrocycles.

We agree with the referee that it is not a rigorous conclusion about the high yield. We have revised the related descriptions in the manuscript as follows:

Abstract part:

“...we report **the high-yield** synthesis of four-, six- and eight-membered tetraphenylethylene (TPE)-based macrocycles on Ag(111) using on-surface reaction. The as-synthesized macrocycles are spontaneously segregated on the surface and self-assemble as **large-area** two-dimensional mono-component supramolecular crystals,...”

Main text:

“... We attribute the multiple **high-yield macrocycle** products in our work to the structural flexibility of the TPE backbone thanks to its non-planar conformation...”

Conclusion part:

“In summary, we demonstrate “one-pot” synthesis of 4-, 6-, and 8-membered TPE macrocycles on a Ag(111) surface **with high yields as the predominant products,**...”

Scheme 1:

6) The authors support their purported high yields by invoking DFT optimized structures which are not shown. I suspect that they have referred to previous work and have neglected to add the citation. However, to establish the reactivity patterns on Ag(111) will require analysis of DFT optimized structures of the compound on that surface. This is essential to help validate the several so far unsubstantiated claims that the authors have made in this case.

Response: We thank this suggestion of the referee. As shown before, we have added the DFT optimized M6/M4/M8 conformer (free-standing) in the revised supplementary materials: Figure S2b for M6, Figure S6 for M4/M8, and Figure S7 for Br₂-TPE. Unfortunately, detailed DFT analysis of the reaction path of the macrocycles including Ag(111) substrate is beyond our computational resource. Knowing this limitation, we explicitly state “Without direct experimental observation or sophisticated theoretical simulation, this intricate process is beyond our comprehension.”

7) *The authors seem keen to emphasize the growth of segregated mono-component domains of the on-surface chemistry products calling it 'phenomenal'. However, I was not especially surprised by this feature since co-deposition of mixtures of molecules often ends at segregated states. One interesting feature of those systems is what occurs at the phase boundaries. Unfortunately, this is not addressed here and only the mixed phase of M5/M7 is shown. Given the chemical similarities between the macrocycles, it could be a point of interest.*

Response: The referee is right that co-deposition of mixtures of molecules can end at segregated states or bimolecular network on a surface. What remarkable here is the huge-size mono-component islands and absence of domain boundaries between the islands, which have not been reported before to our knowledge. Following the referee's comment, we changed "phenomenal" to "very rare."

Main text:

"The growth of such large-area mono-component 2D supramolecular crystals is ~~phenomenal~~ very rare in a multi-component system."

REVIEWERS' COMMENTS:

Reviewer #1 (Remarks to the Author):

The authors have revised the manuscript addressing all of referees' comments. I recommend the publication in Communication Chemistry in its current form.

Reviewer #2 (Remarks to the Author):

The revised manuscript ("On-surface synthesis and spontaneously segregation of conjugated tetraphenylethylene macrocycles") by Li et al. details an on-surface synthesis protocol for the formation of large conjugated macrocycles which are observed to spontaneously segregate into homogenous mono-component domains on the Ag(111) surface. Characterisation is performed via scanning tunnelling microscopy (STM) and the interpretation of the observed structures is supported by density functional theory, DFT, calculations; the details and discussion of the DFT aspects have been significantly improved within the revised manuscript. The manuscript is well presented, and the results shown detail an interesting result which is likely to be significant within the fields of on-surface synthesis and molecular self-assembly.

The revised manuscript addresses the points raised by the reviewers and I recommended the revised manuscript for publication.

Reviewer #3 (Remarks to the Author):

The authors have responded fully and properly to the comments of the reviewers, and also made revisions to the manuscript in line with the comments. The manuscript is now suitable for publication. In the title, 'spontaneously' should be 'spontaneous'.